

# Health risk assessment from habitants of Araró, Michoacán, México, exposed to arsenic by dust, using Monte Carlo probabilistic method

José Leopoldo Mendoza-Lagunas[1,*], Alejandra Damayanti Aguilar-Espinosa[2], Laura Nelly Rodríguez-Cantú[3], Roberto Guerra-González[2], Diana María Meza-Figueroa[4], María Mercedes Meza-Montenegro[5] and Marco A. Martínez-Cinco[2,*]

[1] Dirección Adjunta de Investigación Humanidades y Ciencia, Consejo Nacional de Humanidades Ciencia y Tecnología, Benito Juarez, CIudad de México, Mexico
[2] Facultad de Ingeniería Química, Universidad Michoacana de San Nicolás de Hidalgo, Morelia, Michoacán, Mexico
[3] Facultad de Enfermería, Universidad Michoacana de San Nicolás de Hidalgo, Universidad Michoacana de San Nicolás de Hidalgo, Morelia, Michoacán, Mexico
[4] Departamento de Geología, Universidad de Sonora, Hermosillo, Sonora, Mexico
[5] Departamento de Recursos Naturales, Instituto Tecnológico de Sonora, Obregón, Sonora, Mexico
[*] These authors contributed equally to this work.

Corresponding author
Marco A. Martínez-Cinco,
marco.martinez@umich.mx

## ABSTRACT

Arsenic (As) is a globally distributed metalloid that is emitted from natural sources, including geothermal processes, as well as from anthropogenic activities. The village of Araró, in the state of Michoacán, is located in the Trans-Mexico volcanic belt, which is a highly active geothermal site in central Mexico. The aim of this study is to evaluate the health risk to residents of the town from As exposure from dust through oral, dermal and inhalation pathways, using Monte Carlo simulation. Forty dust samples were randomly collected in Araró village, and these were analyzed using portable X-ray fluoroscopy. The As levels obtained for dust samples ranged from 5.94 to 42.53 mg/kg. Point estimation of hazard quotient (HQ) and its probability distribution was assessed using U.S. Environmental Protection Agency (USEPA) formulas and Monte Carlo simulation, respectively, for oral, dermal and inhalation pathways. Anthropometrical data were obtained from the Health and Nutrition National Survey 2018. Mean average daily dose (ADD) for all age groups (preschooler, Elementary (6–12), adolescent and adult) were below safety limits. A total of 4 and 6% of preschooler and Elementary dermal ADDs were above safety limits. For oral and dermal exposure in children, HQ and hazard index (HI) mean values were higher than other age groups, despite safety limits not being reached. Also, it was found that dermal carcinogenic risk (CR) value for adults may represent a potential cancer risk. Despite a relatively low reported concentration of As it is important that more As exposure routes be explored to determine the severity of the problem because previous studies have shown high As concentrations in drinking water.

## INTRODUCTION

Arsenic (As) is a worldwide distributed metalloid, and its chronic exposure has been highly related with numerous cancer and non-cancer diseases (*Faita et al., 2013*). Several biochemical mechanisms of damage by As have been proposed (Table 1). Arsenic can induce oxidative stress by decreasing activity from glutathione and antioxidant enzymes, even mitochondria become inefficient due to this process (*Fatoki, 2019*; *Firdaus et al., 2018*). Also, As exposure is related to neurological damage by its ability to cross the blood–brain barrier and causes decrease in brain intelligent quotient, alteration in memory performance and neurological transmitters concentration, reduction of synaptic plasticity, signaling, signaling neurogenesis, altered sensory function, peripheral nerve neuropathy and reduced conduction velocity (*Alao et al., 2021*; *Martinez et al., 2008*; *Wang et al., 2013*). Due to its detoxification function, the liver is also affected by As; its presence can compromise hepatic structural function and induce histo-hepatic changes, modifying specific liver enzymes concentrations and functions (*Zhang et al., 2013*; *Zhong et al., 2021*). Renal cells are also susceptible to damage by As; studies had demonstrated that As exposure can increase plasmatic creatinine and urea concentration, that is a renal malfunction indicator (*Palma-Lara et al., 2020*; *Wang et al., 2021*). The mechanism mentioned above can develop symptoms and diseases such colitis, loss of reflexes, weight loss, weakness, anorexia, gastritis, hyperpigmentation, circulatory disorders, lung, liver, renal and skin cancers (*Ditzel et al., 2016*; *Hall, 2002*; *Souza et al., 2018*; *Tokar, Qu & Waalkes, 2011*).

Arsenic can be found in the environment as a result of anthropogenic activities or from natural sources. This metalloid can be used by humans directly as wood preservatives, pesticides and in the car industry, or can also be found as a byproduct from the mining industry (*Chen et al., 2016*). However, not only industry can increase the presence of As; extracting water from wells affects soil's chemistry and can release As to aquifers (*Benner, 2010*). Also, geothermal activity can naturally increase As concentration in environment. The usage of wells in human activities, changes their chemistry increasing metalloid's solubilization in aquifers and thus reaching superficial environment. Ground erosion can also release and transport As that is not naturally available (*Cumbal et al., 2008*). Volcano activity, continental collision zones and continental rifts enhance As concentrations in the Pacific Coast of Latin America because this kind of movements can transport subsoil materials like heavy metals and metalloids. A report in 2012 showed that the range of As in geothermal fluids found in some places of México, Honduras, Guatemala, El Salvador, Costa Rica, Ecuador, Bolivia and Chile was between 0.004–73.6 mg/l (*López et al., 2012*). In México, the presence of As derived from geothermic activity is well documented. A study conducted Near Chihuahua, México, found that rocks in area had around 14% of As content, but they also found a substitution process in which phosphorous substitutes As releasing it to aquifers (*Ren, Rodr & Goodell, 2022*). In other research carried out in Juventino Rosas, Guanajuato, México, *Morales-Arredondo et al. (2016)* found a relationship between high temperatures of aquifers and elevated concentrations of As, 46 µg/l at 48 °C. Its authors also mentioned that, when high As water from aquifers reaches the surface,

**Table 1  As-induced toxicity proposed mechanisms (*Fatoki & Badmus, 2022*).**

| Cellular pathway affected | Effect |
| --- | --- |
| Oxidative stress induction | Mitochondria dysfunction, impaired ATP synthesis, epigenetic changes, DNA damage genetic mutation, resistance to apoptosis, organ toxicity, cancer induction. |
| Bone mineralization decreased | Osteoporosis. |
| Nrf2-keap 1 activation | Cancer cells proliferation and tumorigenesis. |
| Impaired male reproductive function | Infertility. |
| Elevated blood pressure | Cardiovascular disease |
| Hyperglycemia | Diabetes. |
| Lipid metabolism perturbation | Metabolic diseases, dyslipidemias. |

it can also increase As concentration in different matrixes in the environment, like soils (*Morales-Arredondo et al., 2016*).

The exposure to As by drinking water is a well-documented global health problem. According to the World Health Organization (WHO) there are around 220 million people in 50 countries exposed to As (*Podgorski & Berg, 2020*). Principal As exposure routes are ingestion, dermal and inhalation, through contaminated drinking water, food and indoor and outdoor dust (*Arif et al., 2019*). As-contaminated drinking water has been widely studied and is considered a serious public health threat. In addition, recent investigations demonstrated that ingestion of As in soil and dust through hand-to-mouth and object-to-mouth, represent an increased factor of As exposure among children (*Chung, Yu & Hong, 2014*). In 2016 A research conducted in Hubei, China in 2016 authors found that ± 65% of total soil and dust ingested by children came from schools and more than 50% of carcinogenic risk was due to school indoor dust (*Liu et al., 2016*). In 2017 in Cornwall, England, arsenic concentrations reported in dust and soil ranged from 1.7 to 29 mg/kg. The authors also mentioned that arsenic found in these samples was highly bioaccessible and must be widely investigated due to the fact that arsenic can access human cells through dermal exposure, inhalation or ingestion. The last becoming more important because it can enhance As exposure through other sources, like drinking water (*Middleton et al., 2017*).

México has several places with geothermal activity. Michoacán is a state in México, located in the named transverse volcanic system (*Yarza, 2003*). This mountain range goes through Veracruz, Puebla, Michoacán, Tlaxcala, Hidalgo, México, Querétaro, Guanajuato, Guerrero, Jalisco, Colima and Nayarit. In Michoacán these mountains lead to the lakes of Zirahuen, Camecuaro, Chapala, Patzcuaro and Cuitzeo (*Rafidah, Al-Kathiri & Muhammad, 2014*). Araró is located 1 mile away from Cuitzeo lake and 15 miles from Los Azufres geothermic field on the transverse volcanic system (Fig. 1). This is a lacustrine area with thermal springs and agricultural soils (*Suárez-Mota, Téllez-Valdés & Meyer, 2014*). Due to these geothermal characteristics, the surrounding areas in Araró have high concentrations of As in sediments (72.8 mg/kg) and up to 19 g/l in brines (*Birkle & Merkel, 2002*); these conditions increase As exposure and thus increasing health risks for

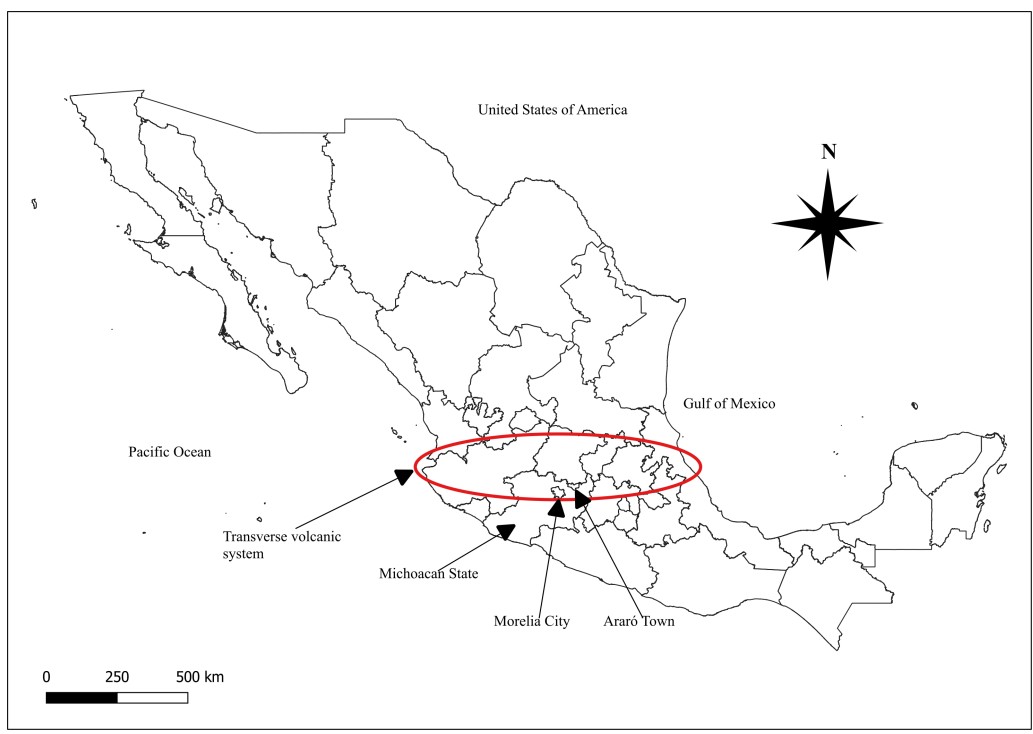

**Figure 1** **Transverse volcanic system geographical location.**

the population. In recent years *Rodriguez-Cantú et al. (2022)* found high As concentrations (25.2 to 66.5 μ/L) in well drinking water of Araró village (*Rodríguez-Cantú et al., 2022*).

Around the world, health risk assessment for As exposure through soil has been widely studied; and the way in which environmental and biological variables interact are shown in Fig. 2. In 2021 in Ban, Iran, *Toolabi et al. (2021)* estimated health risk for children and adults. Results showed that those people were highly exposed to As *via* underground aquifers and dust (13.36 ± 1.51 μg/l); authors emphasize that children were more affected, because of the average of As HI (sum of oral and dermal HQ) was 5.42 ± 0.61 (*Toolabi et al., 2021*). A similar study, conducted in Cornwall, England, reported As concentrations in dust between the range of 1.7 to 29 mg/Kg in dust, even though researchers did not assess the health risk involved, they did conclude that further research is required, due to the fact that home-grown vegetable consumption might increase As intake (*Middleton et al., 2017*). A recent research carried out in San Antonio–El Triunfo mining district, in South Baja California, México, reported HQ values for As *via* urban soil ranged from 0.61 up to 18.9 (*Hernández-Mendiola et al., 2022*). In 2005 a study conducted at Lunda, Angola, reported the presence of heavy metals in urban dust, they mentioned that even though As concentrations and HQ were low (5 μg/g and < 1 respectively) there is a constant exposure to heavy metals, representing a yet unknown health risk specifically for As (*Ferreira-Baptista & De Miguel, 2005*). In recent years (2021) in Zhengzhou, China, heavy metals exposure through dust *via* inhalation and ingestion from different urban areas (commercial, residential, industrial, parks and educational) reported an average

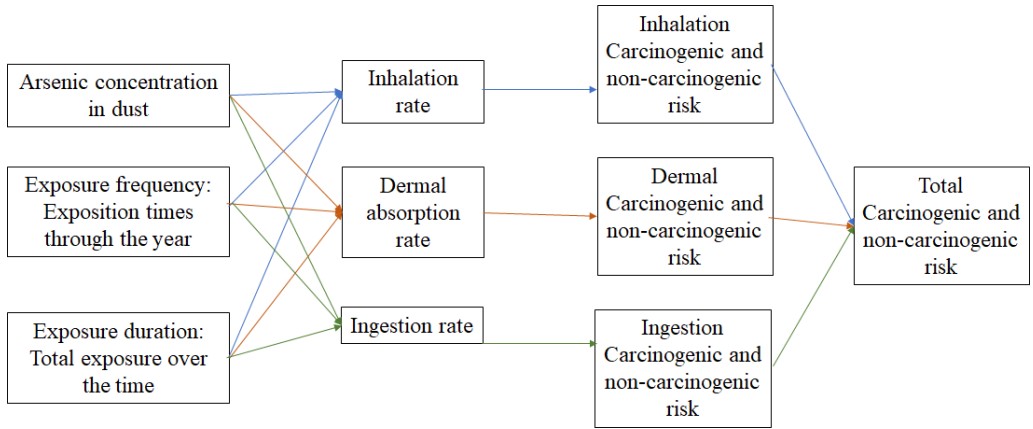

**Figure 2** **Variable interaction in health risk assessment.** Interactions between exposition pathways and exposition routes.

of 11.53 mgAs/Kg. Authors mentioned that Zhengzhou is the 3rd most polluted city in China by As, in industrial and commercial areas HI value increase significantly. In this investigation, HQ and HI results presented were below one, however, researchers insist that, more exposure sources need to be investigated in order to perform an integrated health risk assessment (*Faisal et al., 2021*).

Although the deterministic USEPA method is widely used; using Monte Carlo simulation together enables the creation of many possible scenarios enhancing HQ results; this method decrease results variability and increases its accuracy. *Zuzolo et al. (2020)* evaluated the As distribution and the related probabilistic health risk from many sources, including agricultural soil, grazing soil, stream sediment and water in a whole Italian territory. The population with medium risk (HQ > 1 < 4) lived in Catanzaro (south–west region), Napoli (center region), Trento, Torino, Genova, Bologna and Aosta (north) and high risk (HQ > 4) in Viterbo and Roma (west region). Investigators also mentioned that despite of HQ from ingested soil was relatively low, a significant health risk might be associated with chronic exposure to As, the more susceptible groups, such as children and teenagers, may present non-cancer effects during their lifespan (from birth to an average age of 70 years) (*Zuzolo et al., 2020*). In 2021 a probabilistic multy-pathway health risk was assessed, in this study authors explored exposure to different heavy metals (As, Cd, Cr, Cu, Ni, Pb and Zn) in two possible scenarios (residential and recreational). In 10,000 simulated interactions, they reported that in a residential scenario only 1% of children present HI (sum of HQs) value above one. On the other hand in recreational scenarios, they indicated that 2.5% of children presented a HI value higher than one (*Jiménez-Oyola et al., 2021*).

In Table 2 As concentrations found in different matrices in Michoacán, México are shown. As previously mentioned, Araró is a village located in an active geothermal site, recently performed studies reported high As concentrations in well water, well overexploitation and ground erosion, suggesting chemical changes in subsoil as well as metalloid solubilization and mobilization to surface environment. Also, it is important

**Table 2  As concentrations found in different matrices in Michoacán, México.**

| Autor, year | Matrix | Place | Concentration |
|---|---|---|---|
| Rodríguez-Cantú et al. (2022) | Drinking water | Zinapecuaro, Michoacán, México. | 41.0 µg/L |
| Birkle & Merkel (2000) | Geothermal water | Los Azufres, Michoacán, México. | 5.1 to 24 mg/L |
| Rivas-Valdes et al. (2007) | Dry lake soil | Cuitzeo Lake, Michoacán, México. | 30 to 40 mg/Kg |
| Villalobos-Castañeda et al. (2010) | Sediment river | Lerma river, Michoacán, México. | 0.6 to 6.5 mg/Kg |
| Alarcón-Herrera et al. (2013) | Drinking water | Araró, Michoacán, México. | 0.01 to 63 mg/Kg |
| Osuna-Martínez et al. (2021) | Dry lake soil | Cuitzeo Lake, Michoacán, México. | 0.3–176.1 mg/Kg |
| Israde-Alcantara, Delgado & Chavez (2005) | Municipal dump soil | Morelia, Michoacán, México. | 0.1–0.24 mg/Kg |

to mention that this is the first research of this kind in this community. The aim of this research is to evaluate the health risk of Araro's population to arsenic exposure including ingestion, dermal and inhalation through respiratory pathways, using USEPA formulas and the probabilistic method Monte Carlo simulation.

## MATERIALS & METHODS

### Study area

Araró is a small community ($\pm$11,400 ha) belonging to Zinapecuaro municipality, located in the northern region of the state of Michoacan, 40 km to the northeast of Morelia city and 30 km to the northwest of geothermal hot springs "Los Azufres" (*Sánchez-Núñez et al., 2009*). Araró has lacustrine and agricultural activity with only one water well source for human comsumption. This small town has a lot tourist due to its thermal water springs (*Hiriart Le Bert et al., 2011*).

### Dust sampling and analysis

The number of samples required was calculated according to NMX-AA-132-SCFI-2016 current Mexican legislation, using the following formula.

$n = 2.26xA^{0.31}$;

where:

$n =$ Number of samples

$A =$ Sampling area in hectares.

Once dust samples ($n = 40$) were estimated, they were randomly distributed within a mapped area (Fig. 3) and later taken according current Mexican norm (NMX-AA-132-SCFI-2016). Approximately 500 g of soil was sampled by superficially sweeping with plastic broom and deposited inside of polystyrene bags. All samples were sieved using a rotab equipment with 800, 600, 250, 75 and 45 µm mesh. It is important to mention that after every sieved sample all meshes were cleaned according Mexican regulations. Once obtained 45 µm fractions, samples were analyzed with a portable X-ray fluoroscopy instrument (Thermo Scientific Niton XL3T Analyzer; Thermo Fisher Scientific, Waltham, MA, USA). Analytical quality control (AQC) parameters are shown in Table 3. It is important to mention that equipment calibration was performed by Thermo Scientific, and operational performance was validated according 6200 USEPA method.

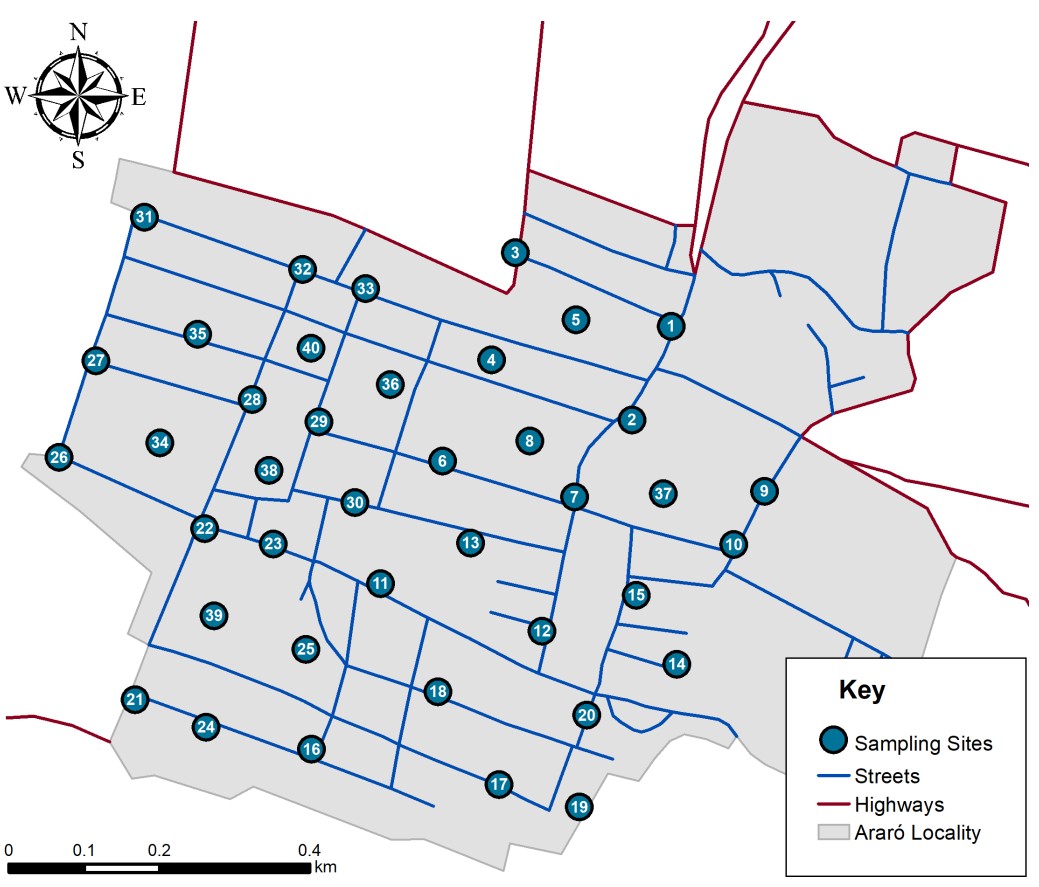

**Figure 3** **Simulation of hazard index distribution in all age groups.**

**Table 3** **Analytical quality control parameters used in current analysis.**

| Parameter | Value |
| --- | --- |
| Variation coefficient (VC) | 0.75% |
| Linearility ($R^2$) | 0.99 |
| Accurancy | 117% |
| Limit of detection | 4 mg/Kg |

## Probabilistic health risk assessment

USEPA formulas were used to compute the risk of different enviromental pathways (Table 4). To calculate average daily dose (ADD) body weight (BW) and age data were obtained from Health and Nutrition National Survey (ENSANUT, 2018), and population was divided by biological age groups (preschoolers, elementary, adolescents and adults) in order to reduce BW and data variability, HQ calculations were performed by using USEPA formulas shown in Table 4. As concentration in dust (AsC) was obtained by analyzing samples. Slope factor (SF), exposure average time (AT), exposure frequency (EF), soil skin adherence factor (AF), event frequency (EV), conversion factor (CF), skin surface area (SA), inhalation rate (InhR), particles emission factor (PEF) and exposure duration (ED)

**Table 4** USEPA formulas used to calculate average daily dose (ADD) hazard quotient (HQ), carcinogenic risk (CR), accumulate carcinogenic risk (aCR), and hazard index (HI).

| Usage | Formula |
|---|---|
| ADD Ingestion (ADDig) | $ADD = \frac{Cs*IR*EF*ED}{BW*AT} * CF$ |
| ADD Inhalation (ADDih) | $ADD = \frac{C*InhR*EF*ED}{PEF*BW*AT}$ |
| ADD Dermal (ADDd) | $ADD = \frac{C*AF*ABS*EV*EF*ED*CF*SA}{BW \quad astAT}$ |
| [1]HQ | $HQ = \frac{ADD}{RfD}$ |
| [2]CR | $CR = \frac{cADD}{SF}$ |
| aCR | $aCR = \sum CR$ |
| HI | $HI = \sum HQ$ |

**Notes.**
[1] A single exposition HQ were calculated used its respectively ADD formula.
[2] For carcinogenic ADD (cADD) same exposition formula is used, just AT variable is changed for cAT.

were obtained from USEPA data base. To perform Monte Carlo simulation, average and range of As concentrations and BW were considered sensitive variables in the calculation of HQ. Risk analyzer add-in for Microsoft Excel was used to perform Monte Carlo simulation; and proper statistical adjustment (as described above) was used for each variable.

## Statistical analysis

Normality tests were performed for AsC in dust, BW, IR, ADD, carcinogenic risk (CR), hazard quotient (HQ) and hazard index (HI). ANOVA method was used to establish differences between means in AsC, utilizing NCSS statistical software V.20.0.8 (2020). Log normal distributions were used in AsC and BW variables respectively, to calculate 10,000 scenarios. In Table 5 calculation parameters descriptions are shown.

## RESULTS AND DISCUSSION

As concentration in $n = 40$ dust samples were in the range of 5.94–42.53 mg/Kg with an average $\pm$ standard deviation (SD) of $15.21 \pm 5.94$ mg/Kg and a median of 12.33 mg/Kg. Sampling sites with the lowest As concentrations were located north of the town. Besides, 13.51% of dust samples were above the values established by the Mexican regulation of 22 mg/Kg (NOM-147-SEMARNAT/SSA1-2004) and were situated south of Araró. Around the world the presence of As in geothermal fluids have been reported, *e.g.*, west USA 7.5 mg/L, México 2.01–6.7 mg/L, Costa Rica 29.13 mg/L, Japan 2.6–9.5 mg/L, Taiwan 4.32 mg/L, Philippines 34 mg/L (*Baba, Uzelli & Sozbilir, 2021*). *Li et al. (2021)* reported 18.84 mgAs/Kg in a district of China; they also collected their samples by sweeping with plastic brooms in random sites around the community. Another finding reported in this study was that in spite of geothermal activity presence the main source of As in dust samples was anthropogenic.

In 2015 in Hermosillo city located in the northwestern region of México, similar concentrations of As in playgrounds ($16.4 \pm 3.5$ mg/Kg), roofs ($15.1 \pm 1.5$ mg/Kg) and roads ($19.3 \pm 4.3$ mg/Kg) were found; this city is located in a natural As presence arid area

**Table 5 Exposure parameters used to calculate non-carcinogenic and carcinogenic risk according USEPA guideline.**

| Parameter | Unit | Parameter characteristics | | Reference |
|---|---|---|---|---|
| Arsenic concentration (AsC) | mg/Kg | Average | 15.6 | This study |
| | | Min–max | 5.94–42.53 | |
| Exposure frequency (EF) | day/year | | 350 | *USEPA (2009)* |
| Exposure duration (ED) | years | Preschooler | 3.51 | *USEPA (2004)* |
| | | Elementary | 8.82 | |
| | | Adolescent | 15.88 | |
| | | Adult | 42.94 | |
| Body weight (BW) | kg | Preschooler | 16.31 | *González-Block et al. (2017)* |
| | | Elementary | 10–39 | |
| | | Adolescent | 33.8 | |
| | | Adult | 14–95.7 | |
| | | | 60.91 | |
| | | | 25.9–158.4 | |
| | | | 72.67 | |
| | | | 45–194.2 | |
| Non carcinogenic Average time (AT) | days | Preschooler | 1,229 | *U.S. Environmental Protection Agency (1988)* |
| | | Elementary | 3,087 | |
| | | Adolescent | 5,558 | |
| | | Adult | 15,029 | |
| Reference dose ingestion (RfD) | mg/kg/day | | 0.0003 | *Faisal et al. (2021)* |
| Ingestion slope factor (SF) | mg/kg/day | | 1.5 | |
| Exposed skin area (SA) | cm2 | Preschooler | 760 | *Department of Health and Aged Care (2012)* |
| | | Elementary | 1,080 | |
| | | Adolescent | 1,840 | |
| | | Adult | 1,935 | |
| Dust to skin adherence factor (SL) | mg/cm2 | | 0.5 | *Department of Health and Aged Care (2012)* |
| Dermal absorption factor (ABS) | | | 0.03 | *USEPA (2001)* |
| Conversion factor (CF) | | | $1 \times 10^{-6}$ | *USEPA (2001)* |
| Dermal contact factor (DFS) | mg * year/kg/day | | 362.4 | *USEPA (2001)* |
| Rate ingestion (Rign) | mg/day | Preschooler | 100 | *EPA & National Center for Environmental Assessment (2017)* |
| | | Elementary | 100 | |
| | | Adolescent | 30 | |
| | | Adult | 30 | |
| Inhalation rate (InhR) | m3/day | Preschooler | 9.5 | *USEPA (2009)* |
| | | Elementary | 12 | |
| | | Adolescent | 15.75 | |
| | | Adult | 14.67 | |
| Particles emission factor (PEF) | m3/kg | | $1.36 \times 109$ | *EPA (1995)* |

and they suggest that the difference in As concentrations was due to source apportionment studies (*García-Rico et al., 2016*). *Morales-Simfors et al. (2020)*, mentioned that México has over 2,300 geothermal sites and almost 25 active volcanos. This geothermal activity may cause interactions between fluids, ground water, rocks and sediments to increase As concentrations in surrounding areas (*Morales-Simfors et al., 2020*). Recently, studies have reported presence of As in sediment (2.24 mg/Kg), geothermal fluid (73.6 mg/L), and groundwater (2.201–49.6 mg/L) through Transmexican Volcanic Belt (*Baba, Uzelli & Sozbilir, 2021*; *Bundschuh et al., 2021*); likewise high As concentrations (3.8 mg/L) were

reported in Araró's hot springs located in northern Michoacán, México (*Morales-Simfors et al., 2020*). This information shows evidence of how aquifers overexploitation and natural geological processes can affect As presence in different regions.

## Non-carcinogenic risk

Table 6 shows ADD data (mean, SD, median, min–max, and P95%) for dermal, oral and inhalation arsenic exposure pathways for all age groups, demonstrating that neither mean nor median exceed their respectively safety criteria (RfD). When maximum values were analyzed individually, it was observed that dermal and oral exposure pathways for 4% of preschooler and dermal pathways for 6% of elementary age groups showed higher values than their respectively background values of RfD. Oral exposure results did not show a significant P95 percentile over than its RfD criteria. It is important to mention that apparently low ADD values does not mean a non-significant arsenic exposure for this population by other arsenic media, due to the fact that in our research we just explored exposure through dust. However in a recent study that evaluated arsenic exposure through drinking water in this same area, it was detected that for elementary age groups had an average ADD ingestion value of $2.31 \times 10^{-03} \pm 8.59 \times 10^{-04}$, which is higher than the RfD criteria (*Rodríguez-Cantú et al., 2022*).

According USEPA, to obtain HQ value, every ADD pathway was divided between RfD data. With this information HQ results are presented in Table 7. It was found that for every single age group no mean or median of every exposure pathway was above safety criteria (HQ = 1). Same behavior was observed with P 95% results. Although some results showed HQ > 1, the amount of simulated data was below 0.001%. Nevertheless, it is important to mention that the sum of all exposure routes for preschooler exhibited values near safety criteria (P 97% = 1.0) In Fig. 4 HI data is shown. HI mean for all age groups were below one, likewise, elementary, adolescent and adult groups didn't present any value above USEPA safety criteria (1). A sensibility variable plot reported by *Rodríguez-Cantú et al. (2022)*, showed that there are three variables that highly influence HI results, being 59% for arsenic concentration, 25% for body weight and 17% for intake rate. This analysis explains why preschooler age groups tend to have higher HI values, since their body weight is lower than the other age groups, their intake rate is higher. Similar results were found in Hubei province, China, where outdoor and indoor dust were analyzed; they found a non-carcinogenic risk below one, even though they only evaluated oral exposure pathway (*Liu et al., 2016*). Another mutly pathway study that was conducted in Northeast China, conlcuded that children's exposure to As was higher than that for adults, due to children playing outdoors (*Xu et al., 2013*). Similarly, Ghanavati and his research group, reported that ingestion is the main exposure pathway for both adults and children, likewise, they reported HI values below one, nevertheless arsenic concentrations found were almost half of what was reported in our investigation (*Ghanavati, Nazarpour & De Vivo, 2019*). In México there has been reports that the main exposure pathway is through ingested dust. A study made in San Luis Potosi, México concluded that HI for children's exposure to As *via* ingestion and inhalation of contaminated dust was above one (*Pérez-Vázquez et al., 2016*). In recent years a highly arsenic contaminated dust site (1,374 mg/kg) was
**Table 6  Average daily dose (ADD) for preschooler, scholar, adolescent and adult (mg/kg/day) exposed to arsenic by dermal, oral and inhalation through dust.**

| Age groups | Pathways | Mean ± SD | Median | Min–Max | P 95% | Safety criteria (RfD) |
|---|---|---|---|---|---|---|
| Preschooler | Dermal | $6.58 \times 10^{-05} \pm 2.83 \times 10^{-05}$ | $6.09 \times 10^{-05}$ | $1.19 \times 10^{-05} - 2.56 \times 10^{-04}$ | $1.18 \times 10^{-04}$ | $1.23 \times 10^{-04}$ |
|  | Oral | $1.05 \times 10^{-04} \pm 4.88 \times 10^{-05}$ | $9.59 \times 10^{-05}$ | $2.07 \times 10^{-05} - 4.31 \times 10^{-04}$ | $1.95 \times 10^{-04}$ | $3.0 \times 10^{-04}$ |
|  | Inhalation | $7.24 \times 10^{-09} \pm 3.35 \times 10^{-09}$ | $6.61 \times 10^{-09}$ | $1.46 \times 10^{-09} - 3.88 \times 10^{-08}$ | $1.32 \times 10^{-08}$ | $3.0 \times 10^{-04}$ |
|  | Sum | $1.70 \times 10^{-04} \pm 5.62 \times 10^{-05}$ | $1.61 \times 10^{-04}$ | $4.45 \times 10^{-05} - 5.12 \times 10^{-04}$ | $2.75 \times 10^{-04}$ |  |
| Schooler | Dermal | $6.75 \times 10^{-05} \pm 3.52 \times 10^{-05}$ | $5.99 \times 10^{-05}$ | $1.29 \times 10^{-05} - 3.61 \times 10^{-04}$ | $1.30 \times 10^{-04}$ | $1.23 \times 10^{-04}$ |
|  | Oral | $5.26 \times 10^{-05} \pm 2.22 \times 10^{-05}$ | $4.89 \times 10^{-05}$ | $9.60 \times 10^{-06} - 2.35 \times 10^{-04}$ | $9.39 \times 10^{-05}$ | $3.0 \times 10^{-04}$ |
|  | Inhalation | $4.63 \times 10^{-09} \pm 2.39 \times 10^{-09}$ | $4.10 \times 10^{-09}$ | $9.35 \times 10^{-10} - 2.56 \times 10^{-08}$ | $9.08 \times 10^{-09}$ | $3.0 \times 10^{-04}$ |
|  | Sum | $1.20 \times 10^{-04} \pm 4.18 \times 10^{-05}$ | $1.13 \times 10^{-04}$ | $3.23 \times 10^{-05} - 4.34 \times 10^{-04}$ | $1.95 \times 10^{-04}$ |  |
| Adolescent | Dermal | $1.16 \times 10^{-05} \pm 4.82 \times 10^{-06}$ | $1.07 \times 10^{-05}$ | $2.45 \times 10^{-06} - 3.80 \times 10^{-05}$ | $2.06 \times 10^{-05}$ | $1.23 \times 10^{-04}$ |
|  | Oral | $8.34 \times 10^{-06} \pm 4.85 \times 10^{-06}$ | $7.39 \times 10^{-06}$ | $4.81 \times 10^{-07} - 5.05 \times 10^{-05}$ | $1.77 \times 10^{-05}$ | $3.0 \times 10^{-04}$ |
|  | Inhalation | $3.20 \times 10^{-09} \pm 1.33 \times 10^{-09}$ | $2.97 \times 10^{-09}$ | $7.39 \times 10^{-10} - 1.14 \times 10^{-08}$ | $5.74 \times 10^{-09}$ | $3.0 \times 10^{-04}$ |
|  | Sum | $1.99 \times 10^{-05} \pm 6.84 \times 10^{-06}$ | $1.90 \times 10^{-05}$ | $4.40 \times 10^{-06} - 6.13 \times 10^{-05}$ | $3.25 \times 10^{-05}$ |  |
| Adult | Dermal | $3.64 \times 10^{-05} \pm 2.02 \times 10^{-05}$ | $3.21 \times 10^{-05}$ | $5.35 \times 10^{-06} - 3.00 \times 10^{-05}$ | $7.30 \times 10^{-05}$ | $1.23 \times 10^{-04}$ |
|  | Oral | $7.36 \times 10^{-06} \pm 5.12 \times 10^{-06}$ | $6.12 \times 10^{-06}$ | $2.17 \times 10^{-07} - 7.66 \times 10^{-05}$ | $1.68 \times 10^{-05}$ | $3.0 \times 10^{-04}$ |
|  | Inhalation | $2.83 \times 10^{-09} \pm 1.56 \times 10^{-09}$ | $2.49 \times 10^{-09}$ | $3.94 \times 10^{-10} - 1.94 \times 10^{-08}$ | $5.69 \times 10^{-09}$ | $3.0 \times 10^{-04}$ |
|  | Sum | $4.38 \times 10^{-05} \pm 2.08 \times 10^{-05}$ | $3.96 \times 10^{-05}$ | $8.23 \times 10^{-06} - 3.06 \times 10^{-04}$ | $8.11 \times 10^{-05}$ |  |

compared with a reference site (5.2 mg/kg) in San Luis Potosi. The Monte Carlo simulation was used to evaluate health risk in children exposed to arsenic *via* this pathway in four communities. It was found that non-carcinogenic data in the high arsenic contaminated site were statistically superior. On the other hand, HQ and HI values found in the reference site showed similar values as ours and HQ and HI results presented were lower, because *Fernández-Macías et al. (2020)* just evaluated dust ingestion pathway.

According to carcinogenic risk through dust exposure (Table 8), it was found that neither oral or inhalations pathways represent a risk, because they are below safety criteria ($1 \times 10^{-05}$). However, it was observed that dermal exposure presented higher values compared to other exposure routes. Elementary, adolescent and adult age groups P95% resulted with CR dermal values above safety criteria for dermal exposure, this result can be attributed to the calculation nature, because according to USEPA guidelines total exposed skin will be lower in preschooler than in other age groups. Even though oral and inhalation pathways alone did not represent a potential cancer risk (based on mean, median and P95%); dermal exposure influence, can increase enough P 95% sum (dermal, oral and inhalation) to exceed safety criteria. Similar results were found by *Ali et al. (2021)*, where urban and rural As exposure for adults and children was assessed. They reported a higher CR in adults ($1.72 \times 10^{-05}$) than in children ($8.77 \times 10^{-06}$); and even higher in rural adults ($2.11 \times 10^{-05}$) and children ($1.07 \times 10^{-05}$) than in urban. Also, they highlight that these CR values can present a potential risk to human health on long-term exposure (*Ali et al., 2021*). As carcinogenic risk has also been reported in México; a study conducted in northern Mexican states, found that women who were exposed to different sources of As (including dust), excreted more As related metabolites, it is important to mention that authors did not calculate a

**Table 7  Hazard quotient (HQ) for preschooler, schooler, adolescent and adult exposed to arsenic by dust: dermal, oral and inhalation pathways.**

| Age groups | Pathways | Mean ± SD | Median | Min–Max | P 95% | Safety criteria |
|---|---|---|---|---|---|---|
| Preschooler | Dermal | 0.2193 ± 0.0945 | 0.2029 | 0.0397–0.8533 | 0.3933 | 1 |
| | Oral | 0.3485 ± 1625 | 0.3187 | 0.0689–1.4389 | 0.6508 | 1 |
| | Inhalation | 0.0005 ± 0.0002 | 0.0004 | 0.0001–0.0026 | 0.0008 | 1 |
| | Sum | 0.5682 ± 0.1875 | 0.5375 | 0.1489–1.7094 | 0.9174 | 1 |
| Elementary | Dermal | 0.2251 ± 0.1172 | 0.1997 | 0.0432–1.2043 | 0.4338 | 1 |
| | Oral | 0.1752 ± 0.0741 | 0.1629 | 0.0320–0.7847 | 0.3130 | 1 |
| | Inhalation | 0.0003 ± 0.0002 | 0.0003 | 0.0001–0.0017 | 0.0006 | 1 |
| | Sum | 0.4005 ± 0.1391 | 0.3771 | 0.1079–1.4477 | 0.6487 | 1 |
| Adolescent | Dermal | 0.0386 ± 0.0161 | 0.0358 | 0.0082–0.1265 | 0.0687 | 1 |
| | Oral | 0.0278 ± 0.0162 | 0.0246 | 0.0016–0.1685 | 0.0590 | 1 |
| | Inhalation | 0.0002 ± 0.0001 | 0.0002 | 0.0001–0.0008 | 0.0003 | 1 |
| | Sum | 0.0667 ± 0.0228 | 0.0634 | 0.0147–0.2043 | 0.1085 | 1 |
| Adult | Dermal | 0.1213 ± 0.0672 | 0.1071 | 0.0178–0.9992 | 0.2435 | 1 |
| | Oral | 0.0245 ± 0.0171 | 0.0204 | 0.0007–0.2553 | 0.0561 | 1 |
| | Inhalation | 0.0002 ± 0.0001 | 0.0002 | 0.0001–0.0013 | 0.0003 | 1 |
| | Sum | 0.1461 ± 0.0693 | 0.1322 | 0.0277–1.0211 | 0.2704 | 1 |

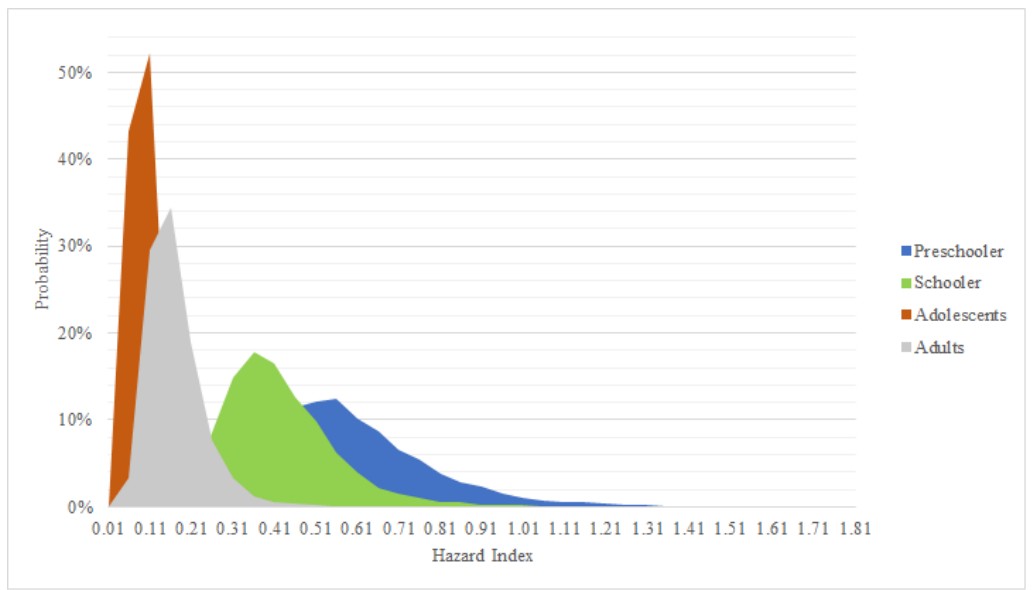

**Figure 4  Simulation for hazard index.** Probabilistic distribution for hazard index in age groups. Safety value = 1.

**Table 8 Carcinogenic risk for As exposure in all age groups.**

| Age groups | Pathways | Mean ± SD | | Median | Min–Max | P 95% | Safety criteria (RfD) |
|---|---|---|---|---|---|---|---|
| Preschooler | Dermal | $2.11 \times 10^{-06}$ | $\pm 9.11 \times 10^{-07}$ | $1.96 \times 10^{-06}$ | $3.83 \times 10^{-07} - 8.26 \times 10^{-06}$ | $3.79 \times 10^{-06}$ | $1 \times 10^{-05}$ |
| | Oral | $3.36 \times 10^{-06}$ | $\pm 1.57 \times 10^{-06}$ | $3.07 \times 10^{-06}$ | $6.64 \times 10^{-07} - 1.39 \times 10^{-05}$ | $6.27 \times 10^{-06}$ | $1 \times 10^{-05}$ |
| | Inhalation | $2.9 \times 10^{-11}$ | $\pm 1.34 \times 10^{-11}$ | $2.65 \times 10^{-11}$ | $5.86 \times 10^{-12} - 1.56 \times 10^{-10}$ | $5.29 \times 10^{-11}$ | $1 \times 10^{-05}$ |
| | Sum | $5.47 \times 10^{-06}$ | $\pm 1.81 \times 10^{-06}$ | $5.18 \times 10^{-06}$ | $1.43 \times 10^{-06} - 5.18 \times 10^{-06}$ | $8.83 \times 10^{-06}$ | $1 \times 10^{-05}$ |
| Elementary | Dermal | $5.45 \times 10^{-06}$ | $\pm 2.84 \times 10^{-06}$ | $4.84 \times 10^{-06}$ | $1.04 \times 10^{-06} - 2.92 \times 10^{-05}$ | $1.05 \times 10^{-05}$ | $1 \times 10^{-05}$ |
| | Oral | $4.42 \times 10^{-06}$ | $\pm 1.87 \times 10^{-06}$ | $4.11 \times 10^{-06}$ | $8.06 \times 10^{-07} - 1.98 \times 10^{-05}$ | $7.89 \times 10^{-06}$ | $1 \times 10^{-05}$ |
| | Inhalation | $4.66 \times 10^{-11}$ | $\pm 2.41 \times 10^{-11}$ | $4.13 \times 10^{-11}$ | $9.42 \times 10^{-12} - 2.57 \times 10^{-10}$ | $9.14 \times 10^{-11}$ | $1 \times 10^{-05}$ |
| | Sum | $9.86 \times 10^{-06}$ | $\pm 3.41 \times 10^{-06}$ | $2.29 \times 10^{-06}$ | $2.65 \times 10^{-06} - 3.54 \times 10^{-05}$ | $1.60 \times 10^{-05}$ | $1 \times 10^{-05}$ |
| Adolescent | Dermal | $5.20 \times 10^{-06}$ | $\pm 2.16 \times 10^{-06}$ | $4.81 \times 10^{-06}$ | $1.10 \times 10^{-06} - 1.70 \times 10^{-05}$ | $9.25 \times 10^{-06}$ | $1 \times 10^{-05}$ |
| | Oral | $1.26 \times 10^{-06}$ | $\pm 7.33 \times 10^{-07}$ | $1.12 \times 10^{-06}$ | $6.27 \times 10^{-08} - 7.64 \times 10^{-06}$ | $2.68 \times 10^{-06}$ | $1 \times 10^{-05}$ |
| | Inhalation | $5.80 \times 10^{-11}$ | $\pm 2.41 \times 10^{-11}$ | $5.38 \times 10^{-11}$ | $1.34 \times 10^{-11} - 2.07 \times 10^{-10}$ | $7.73 \times 10^{-11}$ | $1 \times 10^{-05}$ |
| | Sum | $6.46 \times 10^{-06}$ | $\pm 2.82 \times 10^{-06}$ | $6.10 \times 10^{-06}$ | $1.39 \times 10^{-06} - 1.89 \times 10^{-05}$ | $1.07 \times 10^{-05}$ | $1 \times 10^{-05}$ |
| Adult | Dermal | $1.85 \times 10^{-05}$ | $\pm 1.03 \times 10^{-05}$ | $1.63 \times 10^{-05}$ | $2.72 \times 10^{-06} - 1.52 \times 10^{-04}$ | $3.71 \times 10^{-05}$ | $1 \times 10^{-05}$ |
| | Oral | $3.09 \times 10^{-06}$ | $\pm 2.15 \times 10^{-06}$ | $2.57 \times 10^{-06}$ | $9.13 \times 10^{-08} - 3.22 \times 10^{-05}$ | $7.08 \times 10^{-06}$ | $1 \times 10^{-05}$ |
| | Inhalation | $1.49 \times 10^{-10}$ | $\pm 8.19 \times 10^{-11}$ | $1.31 \times 10^{-10}$ | $2.06 \times 10^{-11} - 1.02 \times 10^{-09}$ | $2.99 \times 10^{-10}$ | $1 \times 10^{-05}$ |
| | Sum | $2.16 \times 10^{-05}$ | $\pm 1.05 \times 10^{-05}$ | $1.95 \times 10^{-05}$ | $4.14 \times 10^{-06} - 1.55 \times 10^{-04}$ | $4.03 \times 10^{-05}$ | $1 \times 10^{-05}$ |

CR for study population, nevertheless, they mentioned that the increase in As metabolites excretion (specially MMA) is associated with cancer incidence (*Gamboa-Loira, Cebrián & López-Carrillo, 2020*). Around Mexican transverse volcanic system, similar results have been reported. In 2017, Castro-Gonzalez and his research group assessed the health risk in inhabitants from Tlaxcala and Puebla, México. Due to the low As concentration in soil, they found a low HQ for both children and adults. In spite of this findings adults showed higher values for CR than children, they even mentioned that dermic exposure pathway increase CR in adults (*Castro-González et al., 2017*).

## CONCLUSIONS

Arsenic concentrations found in dust samples from Araró, Michoacán, México were below Mexican regulations. In addition, HQ and HI results obtained from Monte Carlo probabilistic method did not show any health issues. This assumption is supported by P 95% reported in every single age group (preschooler, elementary, adolescent and adult). Nevertheless, it was found that because of their lower body weight and higher As rate intake for oral and dermal pathways, preschooler and elementary HQ and HI tend to be higher than other age groups without reaching safety criteria. In addition, it was observed that in spite of a relatively low As exposure through dust by dermal, ingested and inhaled pathways, CR value showed potential cancer risk for adults in dermal exposure. As exposure is a serious health problem in this area; previously, our research group reported out of regulation As concentrations in drinking water in Zinapecuaro de Figueroa municipality, this means that people who live in the surrounding areas are exposed to As *via* drinking

water and dust. It is important to investigate more exposure routes, in order to visualize the As problems in the Araró geothermal area.

### Funding
This work was supported by the Universidad Michoacana de San Nicolás de Hidalgo and Consejo Nacional de Humanidades Ciencia y Tecnología. The funders had no role in study design, data collection and analysis, decision to publish, or preparation of the manuscript.

### Grant Disclosures
The following grant information was disclosed by the authors:
Universidad Michoacana de San Nicolás de Hidalgo and Consejo Nacional de Humanidades Ciencia y Tecnología.

### Competing Interests
The authors declare there are no competing interests.

### Author Contributions
- José Leopoldo Mendoza-Lagunas conceived and designed the experiments, performed the experiments, analyzed the data, prepared figures and/or tables, authored or reviewed drafts of the article, and approved the final draft.
- Alejandra Damayanti Aguilar-Espinosa conceived and designed the experiments, performed the experiments, analyzed the data, prepared figures and/or tables, authored or reviewed drafts of the article, and approved the final draft.
- Laura Nelly Rodríguez-Cantú conceived and designed the experiments, analyzed the data, authored or reviewed drafts of the article, and approved the final draft.
- Roberto Guerra-González conceived and designed the experiments, performed the experiments, authored or reviewed drafts of the article, and approved the final draft.
- Diana María Meza-Figueroa analyzed the data, authored or reviewed drafts of the article, and approved the final draft.
- María Mercedes Meza-Montenegro analyzed the data, prepared figures and/or tables, authored or reviewed drafts of the article, and approved the final draft.
- Marco A. Martínez-Cinco conceived and designed the experiments, performed the experiments, analyzed the data, prepared figures and/or tables, authored or reviewed drafts of the article, and approved the final draft.

### Data Availability
The raw data are available in the Supplementary Files.

### Supplemental Information
Supplemental information for this article can be found online at http://dx.doi.org/10.7717/peerj.18805#supplemental-information.

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
