# Peer review of "Health risk assessment from habitants of Araró, Michoacán, México, exposed to arsenic by dust, using Monte Carlo probabilistic method"

_PeerJ, doi:10.7717/peerj.18805_

## Round 0.1 · original submission · Major Revisions

I agree with the reviewers that this paper is well-organized overall, and contains useful information. However, there are some issues, including both theoretical concerns and aspects of the written paper, which would need to be resolved before the work can be considered for publication. Multiple reviewers pointed out the relatively low quality of the figures, which will need to be reproduced at higher resolution. In addition, there is a tendency to switch between scientific notation and decimal notation in the Tables. Unless there is a compelling need to alternate formats, it would be helpful if the tables were consistent. Each of the reviewers also called out specific concerns which will need to be addressed if a revised version is submitted. In particular, the concern regarding sample size (40 dust samples) will need to be justified, and the methodological gaps addressed in the next version. Finally, while I am sympathetic to the difficulty in publishing work in a non-native language, there are some persistent errors of English grammar and vocabulary which affect the readability of the current manuscript. For instance, the word "exposition" is often used where the context suggests that "exposure" is the correct term, among other minor errors. I would strongly suggest that the next revision be carefully reviewed by a native English speaker in order to minimize confusion.
I hope and expect that you will find the reviewers' comments constructive, and that you will consider carefully the points that they have raised. I believe that if these issues are resolved, this work will be suitable for publication in a revised version. Please note that any revision submitted will be sent to the original reviewers (and possibly other expert reviewers if needed) to ensure that their concerns are suitably addressed. Thank you again for submitting your work to PeerJ, and I look forward to reading the revised submission.

Reviewer 1 ·

Basic reporting

This manuscript uses Monte Carlo to evaluate the health risk of arsenic in dust for Araro Mexico. The English is clear and unambiguous and introduction is comprehensive.

But some minor grammar mistakes are found. For example, in line 178-179: according to.

However, there are too many display items in the main text, including 8 tables and 3 figures. The figure resolution is poor. I suggest to put some to supporting information and present some tables in appropriate figure such as violin/box plot with significance symbol.

Experimental design

The Monte Carlo simulation part needs to include the details of parameters for the software Add-ins to compute the 10,000 repetitions.

Also, is 40 samples enough?

In line 182, please cite the method.

In line 185, it is better to present the formula with formula label instead of putting them in a table.

Validity of the findings

The findings are well stated but the detailed methodology parameters are lacked, such that it is hard to evaluate the reliability and robustness of the results at this moment.

Reviewer 2 ·

Basic reporting

Although the topic is of interest to the scientific community, before considering it for publication, this paper should be improved. Authors should reconsider the main objective of the paper according to the content. They should try to synthesize and emphasize the study's main findings and avoid long sentences. Furthermore, authors should avoid drawing risky conclusions.
Evaluation; Minor Revision.
1. Abstract; The authors should be revised the abstract, it is too general. Moreover, it could be further developed, there is a lot of interesting data in the article. An informative and representative conclusion should be added to the abstract.

Experimental design

2. The environmental parameters used in the analysis are clearly written, and it seems that a diagram showing the relationship between the parameters and the independent and dependent variables is needed.
3. In the main text, many numeric data are given with too many significant figures; 2 significant figures suffice, and 3 suffice in case the first significant figure is "1".
4. Line 211; Please specify NCSS 2020 software program.

Validity of the findings

5. If As is Non-carcinogenic risk, Why As is will use to study in this research?
6. You must provide all the figures in high resolution. Make all the labels and legends more legible.
7. Conclusion; the findings could be further developed, there is a lot of interesting data in the article.

Additional comments

-

Annotated reviews are not available for download in order to protect the identity of reviewers who chose to remain anonymous.

Reviewer 3 ·

Basic reporting

On the abstract, the explanation of arsenic (i.e. the characteristic and health effect) should delete on this section. Then, you can provide or add this information on the first paragraph of the introduction.

Experimental design

1. Can you add the intext citation for the 6200 USEPA method, USEPA database, and the method for ANOVA?

2. To easily understanding, you should summary your methodologies as the diagram or picture.

Validity of the findings

-

Additional comments

-

---

## Round 0.2 · accepted · Accept

Thank you for your careful attention to the issues raised by reviewers of the previous manuscript. I am pleased to inform you that the revised version is now suitable for publication, and I look forward to the release of this work. Thank you again for choosing PeerJ as a forum for your research.

Reviewer 1 ·

Basic reporting

The revised manuscript conveys clear and unambiguous professional English used throughout.

Experimental design

The revised manuscripts have appended the necessary methodology details with rigorous criteria.

Validity of the findings

The revised manuscript has well stated conclusion sections after the methodology has been detailed.

Reviewer 2 ·

Basic reporting

This revised version is suitable for publication.

Experimental design

-

Validity of the findings

-

Additional comments

-

Reviewer 3 ·

Basic reporting

No comment

Experimental design

no comment

Validity of the findings

No comment